# A dynamic foot model for predictive simulations of human gait reveals causal relations between foot structure and whole-body mechanics

**Lars D'Hondt**[1]*, **Friedl De Groote**[1], **Maarten Afschrift**[2]

**1** Department of Movement Sciences, Katholieke Universiteit Leuven, Leuven, Belgium, **2** Department of Human Movement Sciences, Vrije Universiteit, Amsterdam, The Netherlands

* lars.dhondt@kuleuven.be

**Data Availability Statement:** All models and code are available at https://github.com/Lars-DHondt-KUL/3dpredictsim/tree/four-segment_foot_model.

## Abstract

The unique structure of the human foot is seen as a crucial adaptation for bipedalism. The foot's arched shape enables stiffening the foot to withstand high loads when pushing off, without compromising foot flexibility. Experimental studies demonstrated that manipulating foot stiffness has considerable effects on gait. In clinical practice, altered foot structure is associated with pathological gait. Yet, experimentally manipulating individual foot properties (e.g. arch height or tendon and ligament stiffness) is hard and therefore our understanding of how foot structure influences gait mechanics is still limited. Predictive simulations are a powerful tool to explore causal relationships between musculoskeletal properties and whole-body gait. However, musculoskeletal models used in three-dimensional predictive simulations assume a rigid foot arch, limiting their use for studying how foot structure influences three-dimensional gait mechanics. Here, we developed a four-segment foot model with a longitudinal arch for use in predictive simulations. We identified three properties of the ankle-foot complex that are important to capture ankle and knee kinematics, soleus activation, and ankle power of healthy adults: (1) compliant Achilles tendon, (2) stiff heel pad, (3) the ability to stiffen the foot. The latter requires sufficient arch height and contributions of plantar fascia, and intrinsic and extrinsic foot muscles. A reduced ability to stiffen the foot results in walking patterns with reduced push-off power. Simulations based on our model also captured the effects of walking with anaesthetised intrinsic foot muscles or an insole limiting arch compression. The ability to reproduce these different experiments indicates that our foot model captures the main mechanical properties of the foot. The presented four-segment foot model is a potentially powerful tool to study the relationship between foot properties and gait mechanics and energetics in health and disease.

## Author summary

During every step, the foot absorbs the shock as it strikes the ground, supports the body weight, and transfers calf muscle forces to the ground to push off towards the next step.

**Funding:** This work was supported by KU Leuven (https://www.kuleuven.be/kuleuven, Internal Funds C24M/19/064 to FDG); Research Foundation Flanders (FWO, https://www.fwo.be/, junior research project fundamental research G0B4222N to FDG) and FWO postdoctoral fellowship (12ZP120N to MA). The funders had no role in study design, data collection and analysis, decision to publish, or preparation of the manuscript.

**Competing interests:** The authors have declared that no competing interests exist.

Although experiments have shown that foot structure affects both foot and whole-body movement, a deep understanding of the role of different muscles, ligaments, and other tissues in the foot is still lacking. Examining the relation between alterations in individual foot properties, e.g. muscle strength or arch height, and walking is challenging in experiments, but can be done in model-based simulations. Yet, existing models that are suitable for simulating walking oversimplify the foot. Here, we present a new, more detailed foot model. Simulations with our model captured many features of human walking not captured by simulations with simple models. We found that the stiffness of the foot-ankle complex and especially the ability to stiffen the foot had a large effect on the walking pattern. A reduced ability to stiffen the foot resulted in simulated walking patterns resembling those observed in patients with flexible feet. In the future, we will use our model to investigate how foot deformities alter walking in patients and to design personalized treatments.

## Introduction

During locomotion, we interact with the environment through our feet; in addition to weight-bearing, they transfer forces needed to accelerate and decelerate the body during movements. The foot's arched shape enables stiffening the foot to withstand high loads, without compromising the flexibility of the foot [1]. A compliant foot enables the storage and release of elastic energy in the ligaments crossing foot joints, which might contribute to gait efficiency [2]. Intrinsic foot muscles can modulate the visco-elastic properties of the foot to absorb a shock, assist with push-off, or create a stiff lever to transfer force [3,4]. Experimental studies have shown that altering foot properties through custom insoles [5,6] or local anaesthesia of intrinsic muscles [7] had significant effects on ankle-foot and whole-body mechanics and energetics during walking [5–7]. Yet, as it is hard to manipulate individual structures (muscles, tendons, ligaments, bones) experimentally, probing their contributions to gait mechanics and energetics has been difficult. Predictive simulations based on a musculoskeletal model provide an alternative approach to studying how foot structure shapes whole-body gait. Predictive simulations generate de novo gait patterns based on a musculoskeletal model. Because they do not rely on experimental gait data, changes in the predicted gait can be directly attributed to changes in the model. However, using predictive simulations to study the relationship between foot properties and gait mechanics requires sufficiently detailed foot models. So far, foot models used in predictive simulations have been very simple, especially in light of the complex foot anatomy. Three-dimensional simulations with multi-segment foot models have considered up to three segments (talus, hindfoot-midfoot-forefoot, and toes) [8–13], but have not considered a deformable foot arch. Here, we present a novel four-segment (talus, calcaneus, midfoot-metatarsals, toes) musculoskeletal model of the foot that can be used with predictive gait simulations and use this model to study how different foot structures (the longitudinal arch, plantar fascia, plantar fat pads, and muscles actuating the foot) influence simulated whole-body movement.

The compliant longitudinal foot arch has been suggested to improve gait efficiency by storing and releasing elastic energy [2]. The foot absorbs energy while it is flat on the ground and the body's centre of mass moves forward over the foot. Releasing the stored energy at push-off is believed to enable efficient generation of high push-off power [2,6]. Energy is mainly stored by deformation of the medial longitudinal arch [2,6]. Controlled loading of cadaveric feet showed that plantar fascia, long and short plantar ligaments, and spring ligaments are all

important in the energy-storing mechanism [2]. Although elastic energy is stored in the foot during both walking and running, this energy storage seems to contribute more to running than to walking efficiency [2,6]. Reducing arch compression with an insole resulted in a decrease in elastic energy returned from the arch in both walking and running, but changes in metabolic cost of transport were only observed in running [6]. We modelled a compliant longitudinal foot arch to gain more insight into its contribution to energy storage and release in the foot.

During walking, most push-off power comes from the triceps surae and Achilles tendon [14]. While the body moves over the standing foot, energy is stored in the Achilles tendon which rapidly releases the stored energy to generate a powerful forefoot push-off [1,14]. This process is facilitated by the twisted morphology of the Achilles tendon, enabling it to endure large strains and store a substantial amount of energy compared to other tendons [15]. To use this energy effectively during push-off, the foot must function as a stiff lever to minimise energy dissipation [16,17]. Hence, foot stiffness influences the propulsive power generated around the ankle [5,7]. Therefore, we investigated how Achilles tendon stiffness influenced simulated gait mechanics using a foot model with a compliant longitudinal arch.

It has been suggested that the windlass mechanism stiffens the foot at push-off. Hicks described how extending the toes tensions the plantar fascia by winding it around the metatarsal heads [17]. This then pulls the base of the medial longitudinal arch together, raising the arch and stiffening the foot [17]. However, to be effective, the windlass mechanism requires the plantar fascia to be very stiff, which would hinder its ability to store elastic energy [18]. During vertical loading of a standing foot, stiffness of the foot arch did not increase with the toes being more extended [18,19]. Hence, it is unclear whether the windlass mechanism contributes to stiffening the foot. We modelled the plantar fascia to study whether the windlass mechanism contributes to stiffening the foot at push-off.

Intrinsic foot muscles can modulate foot stiffness and might therefore play an important role in optimising efficiency by enabling the foot to function both as an elastic structure that stores and releases energy and as a stiff lever. The intrinsic muscles in the superficial layers of the plantar aspect of the foot have short fibres, a high pennation angle and a long tendon [20,21], which enables these muscles to produce force to tension their tendon to absorb and return elastic energy in a controlled way [22]. Multiple plantar intrinsic muscles originate on the calcaneus and insert into different toe bones [20,21]. Thus contraction of these muscles will add to the stiffness of the foot in parallel with the stiffness provided by the plantar fascia [4]. Electromyographic recordings have demonstrated that intrinsic foot muscles are active during the stance phase of walking [4,23,24]. This activity is highest in the second half of the stance phase when the triceps surae contract and the ankle plantarflexion moment is high [4,23]. Preventing intrinsic muscle contraction by injecting a nerve block decreased the push-off power of the ankle-foot complex, yet arch deformation and cost of transport were unaffected [7]. We modelled intrinsic foot muscles to elicit their contribution to foot stiffness and walking mechanics.

Foot stiffness also depends on the mechanical properties of the plantar fat pads. Especially at initial contact when only the heel is in contact with the ground, the heel pad might be the main contributor to foot stiffness. The visco-elastic fat pad under the heel has been shown to absorb the shock and decelerate the lower leg at initial contact [1]. Therefore, we also explored the stiffness of the foot-ground contact.

Predictive simulations based on simpler foot models than the one proposed here have already shown that foot structure affects whole-body movement. Modelling a compliant foot arch and toes in 2D gait simulations increased the cost of transport, especially for lower gait speeds [25]. Increasing the stiffness of the longitudinal foot arch resulted in a lower cost of

transport, both in simulations and experiments [26]. However, insights based on 2D models are limited given that gait is a 3D movement and that there is considerable out-of-plane movement at the ankle and foot (e.g. ankle inversion/eversion). In 3D simulations, modelling the toes as a separate segment mainly improved the accuracy of predicted knee kinematics, kinetics, and vasti activity during stance [8]. While simulations based on detailed 3D musculoskeletal models already capture many key features of human walking mechanics and energetics, predicting realistic ankle mechanics remains challenging. Simulations do not capture ankle dorsiflexion during stance [8,12,27], or underestimate ankle plantarflexion in terminal stance [8,10,11,28–30]. They capture the peak in soleus activation during late stance but fail to predict the early onset and slow increase of soleus activation [8,9,11,13,27–31]. Reducing the Achilles tendon stiffness (relative to the default Hill-type muscle model [32]) resulted in an earlier onset of soleus activation and a small increase in stance ankle dorsiflexion, yet ankle kinematics remained inaccurate [8]. Because foot mechanisms, such as the compliant longitudinal arch and the windlass mechanism, are thought to support ankle push-off, we expected that using a more detailed foot model would result in more accurate predictions of ankle mechanics.

Detailed foot models available in the literature are not directly suitable for use in predictive simulation. These models have mainly been developed for use in data-driven analyses, i.e. analyses that used experimental data as input (e.g. marker trajectories, electromyography, or ground reaction forces measured during gait) to infer information about variables that could not be measured (e.g. motion of individual bones, or muscle forces). These multi-segment foot models consider 3–26 foot segments, with up to 35 degrees of freedom [33–37]. Existing models include up to 30 intrinsic foot muscles and up to 66 ankle-foot ligaments [34,35]. Depending on their intended use, the models include inertial properties of the segments, and/or soft tissues (e.g. muscle-tendon units, ligaments). It is not straightforward to use these detailed models for predictive gait simulations that generate de novo movement patterns because they have only been validated to estimate the kinematics of individual bones [33–35]. It remains unclear how accurate the estimated muscle and ligament forces are, and such validation is challenging given the high level of detail. We chose to gradually increase the complexity of the foot model starting from the state of the art in predictive simulations and the open questions in foot function outlined above. This incremental approach makes it more tractable to analyse how different modelling choices (e.g. adding a degree of freedom or selecting a value for a muscle parameter) affect the predicted gait. Our developments were in part informed by a foot model designed for inverse analyses, and we believe that such models will continue to inspire models for predictive simulations.

Here we present a dynamic four-segment foot model (talus, hindfoot, midfoot-forefoot, and toes) for 3D predictive simulations of gait. The model describes the passive stiffness of the midtarsal joint and includes the plantar fascia. The foot is actuated by extrinsic as well as intrinsic foot muscles. We used our model to simulate different conditions. First, we simulated how a standing foot deformed under vertical loading (cfr. experiments [2,18,19]) to evaluate whether our model captured the mechanical properties of the foot. Second, we simulated walking for a healthy adult based on a whole-body model including our new foot model to evaluate how our foot model influenced whole-body kinematics, kinetics, energetics, and muscle activation patterns. We evaluated simulations against experimental data and compared them with state-of-the-art simulations based on a three-segment foot [8]. Third, we simulated walking based on a series of foot models with alternative properties to evaluate the effects of modelling choices and to study the contribution of individual foot structures to whole-body gait. Fourth, we simulated experiments that manipulated foot properties during gait. We evaluated how limiting foot arch compression affects the metabolic cost of walking and running (cfr. [6]),

and how local anaesthesia of intrinsic foot muscles affects walking kinematics and energetics (cfr. [7]). We identified three properties of the ankle-foot complex that are important to capture ankle and knee kinematics, soleus activation, and ankle power of healthy adults: 1) compliant Achilles tendon, 2) stiff heel pad, 3) the ability to stiffen the foot. The latter requires sufficient arch height and contributions of plantar fascia, and intrinsic and extrinsic foot muscles. A reduced ability to stiffen the foot results in walking patterns with reduced push-off power.

## Material and methods

### Musculoskeletal model

We adapted the musculoskeletal model used by Falisse et al. [8]–which is based on OpenSim's gait2392 model [38,39] and the model proposed by Hamner et al. [40]–to include a midtarsal joint, plantar fascia, and a plantar intrinsic foot muscle. The resulting model has 33 skeletal degrees of freedom (dofs) (pelvis as floating base: 6 dofs, hip: 3 dofs, knee: 1 dof, ankle: 1 dof, subtalar: 1 dof, midtarsal: 1 dof, metatarsophalangeal (MTP): 1 dof, lumbar: 3 dofs, shoulder: 3 dofs, and elbow: 1 dof). We used Newtonian rigid body dynamics to model skeletal motion [41,42]. The lower limb and lumbar joints are actuated by 94 Hill-type muscle-tendon units (92 muscles according to the gait2392 model and the right and left plantar intrinsic foot muscles as described below) [32,43]. Muscle excitation-activation coupling is given by Raasch's model [44]. The shoulder and elbow joints are actuated by ideal torque actuators [8]. Each joint has viscous friction (coefficient 0.1 Nm s rad$^{-1}$) and nonlinear stiffness to represent the net effects of ligaments [11,45].

A first series of model adaptations aimed at better capturing the properties of muscles spanning the ankle (Table A in S1 Text). During maximal isometric contraction of the triceps surae, Achilles tendon strain is reported to be 5.6–8% [46,47]. We reduced the normalised stiffness of the Achilles tendon such that maximal isometric contraction resulted in strains in the experimental range (6.8%). To better capture experimental peak isometric ankle moments, we increased the maximal isometric force of the triceps surae by 20% with respect to the force of the gait2392 model [38,39] (Fig E in S1 Text). We adapted the parameters related to the passive stiffness of the muscles spanning the ankle joint to better capture the experimental passive moment-angle relationship of the ankle. Passive stiffness of all muscles crossing the ankle joint was increased by shifting their passive force-length characteristic [32] by 10% of the optimal fibre length towards smaller fibre lengths. The resulting passive ankle moments are consistent with measurements [48] (see section 1 in S1 Text for additional information, validation, and sensitivity analysis).

We modelled the foot arch as a midtarsal joint with a single rotational degree of freedom, connecting the calcaneus and midfoot segments. Midfoot and forefoot segments are rigidly connected. Midtarsal joint centre, segment definitions, and segment mass properties were taken from Malaquias et al. [35]. The orientation of the midtarsal joint axis was set according to the mean finite helical axis calculated from the motion capture data during the stance phase of walking using the algorithm provided by Ancillao [49] (See section 2 in S1 Text for the sensitivity of midtarsal joint axis orientation). We defined the rotational stiffness of the midtarsal joint by combining the contributions of long and short plantar, and spring ligaments [35], considering a nonlinear stress-strain characteristic [50] (section 3 in in S1 Text).

We modelled the plantar fascia as a single elastic element with origin on the calcaneus and insertion on the toes [35], which wraps around a cylinder with radius 9.5 mm at the metatarsal head [51]. Based on in vivo ultrasound imaging in healthy adults, we set a nominal plantar fascia cross-section of 70 mm$^2$ [52]. We assumed a purely elastic plantar fascia with a nonlinear

stress-strain characteristic presented by Natali et al. [53]. This stress-strain characteristic was based on uniaxial tension tests of plantar fascia samples [53,54]. We alternatively considered using the plantar fascia stress-strain characteristic presented by Gefen [50], however this stress-strain curve was more compliant than experimentally observed (Fig L in S1 Text). To test the effect of additional plantar fascia stiffnesses, we combined the stress-strain characteristic from Natali et al. with larger cross-sections (210 mm$^2$, 140 mm$^2$), and with a smaller cross-section (24 mm$^2$, cfr. cadavers [55–57]) (Fig M in S1 Text). Plantar fascia slack length (146 mm) was chosen such that a standing foot, bearing only the weight of the foot and tibia, is in the anatomical position.

We modelled the plantar intrinsic muscles of a foot as a single Hill-type muscle because they act as a functional unit [24]. Origin, insertion, and wrapping of the muscle are the same as the plantar fascia. Parameters describing the muscle were derived from cadaveric measurements of flexor digitorum brevis (FDB), abductor hallucis (ABDH), quadratus plantae (QP), and abductor digiti minimi (ABDM), which all span the longitudinal foot arch and MTP joint [20,21]. Based on the mean fibre lengths reported by [21], we set the optimal fibre length to 23 mm. Maximal contraction velocity is 10 optimal fibre lengths per second [11]. The pennation angle at optimal fibre length is 20˚, equal to measurements of FDB and AH [20]. To determine the physiological cross-sectional area (PCSA) of the lumped muscle, we summed the PCSA of FDB, ABDH, QP, and ABDM for a total of 1831 mm$^2$ [21]. Since the PCSA is obtained from donors with an average age of 71 years, we compensated for age-related muscle atrophy. Older adults show a 24–40% reduction in strength in ankle and foot muscles, in comparison to young adults [58]. Assuming a 24% decrease in PCSA due to ageing, the lumped PCSA is 2400 mm$^2$ for a young adult. With a specific tension of 25 N cm$^{-2}$ [59], the maximal isometric force is 600 N. To model the stiffness provided by the internal tendon in highly pennate muscles, we shifted the passive force-length characteristic [32] with 10% of the optimal fibre length towards shorter fibre lengths. We determined tendon slack length based on the assumption that muscle fibres work close to their optimal length during walking. Since we modelled the muscle-tendon unit along the same path as the plantar fascia, it has the same length as the plantar fascia. The plantar fascia is expected to have 0–4% strain while walking [54,60] allowing us to estimate the expected muscle-tendon length based on the selected plantar fascia slack length. We then calculated normalized fibre lengths for different tendon slack lengths assuming an isometric contraction at an activation of 0.5. We selected the tendon slack length (123 mm) that resulted in fibre lengths around the optimal fibre length. To test the influence of parameter choices, we repeated the walking simulations with different parameter values: we tested maximal isometric force values of 120 N, 300 N, 456 N (i.e. not compensating for age-related atrophy), 720 N, 900 N and 1200 N. We tested optimal fibre lengths of 23±4 mm and 23±8 mm (4 mm standard deviation on reported FDB fibre length [21]). We tested tendon slack lengths of 121 mm, 122 mm, 124 mm and 125 mm (section 5 in S1 Text). We also tested a model without plantar intrinsic muscles.

The model was scaled to the anthropometry of the subject (see below) with the OpenSim scale tool [38,39]. Scale factors were calculated based on marker data of the static trial. For the foot (talus and distal segments), we included manual measurements to determine non-uniform scale factors. Anteroposterior scaling was based on the distance between the dorsal calcaneus marker and the hallux marker. Mediolateral scaling was based on the distance between the markers on the heads of the first and fifth metatarsal (section 10 in in S1 Text). In the vertical direction, the vertical distance between the centres of the medial malleolus and the first metatarsal head (based on manual measurements) determined the size of the foot. Because the scaling of the foot was non-uniform, insertion points of the tibialis anterior and posterior, and peroneus (brevis, longus, and tertius) were distorted. We substituted these insertion point

locations from an alternative musculoskeletal model [35], after scaling it to match our subject-specific model. Foot proportions of this alternative model were close to the subject's foot proportions; thus, scaling did not cause distortions. The optimal fibre length and tendon slack length of the adjusted muscles were linearly scaled (cfr. the OpenSim scaling method for these parameters [38,39]). To test the effects of this scaling method, we also created a model with uniformly scaled feet. The uniform scale factor is proportional to the distance between the dorsal calcaneus marker and the hallux marker. Uniform scaling resulted in a lower foot arch compared to non-uniform scaling.

We modelled foot-ground contact as deformable spheres (foot) interacting with a rigid plane (ground). The force acting on each sphere was calculated from Hertzian stiffness and Hunt-Crossley dissipation [61,62]. Each foot has five contact spheres: heel pad, lateral longitudinal arch, ball of the foot (2 spheres), and toes (Fig 1). The location and radius of each sphere were chosen to represent areas with high plantar pressure during standing and walking [63,64], and based on anatomical reference [65]. To tune the vertical position of the contact spheres, we tracked experimental joint angles from the static trial with skeletal and contact dynamics. We solved the static force equilibrium and adjusted the vertical contact positions to reduce offset with respect to experimental pelvis-to-ground coordinates. We set the stiffness parameter of each contact sphere to 10 MPa, instead of the 1 MPa used previously [8,11]. Preliminary simulations showed that this improved the realism of the contact deformation during walking. We evaluated how the configuration and stiffness of the contact spheres influenced the predicted gait (section 6 in S1 Text).

To dissociate the effect of modelling the foot in more detail versus altering parameters, we created a nominal 3-segment foot model that is equivalent to our nominal 4-segment model with the midtarsal joint locked in anatomical position. To be consistent with the model from Falisse et al., the MTP joint of the 3-segment model was not actuated by muscles, but by a spring (25 Nm/rad) and damper (2 Nm s/rad) [8]. We tested an alternative 3-segment foot model with muscle-actuated MTP (section 7 in S1 Text).

### Simulations of standing foot under vertical loading

We replicated a series of experiments that evaluated the deformation of the foot under external loading [2,18,19] in simulation and compared simulated and experimental outcomes to validate our model. To simulate these experiments, we considered only the right tibia and foot of our musculoskeletal model. The angle between toes and ground was fixed (cfr. [18,19]), and muscle activity was set to zero for cadaver experiments and to a small baseline activation (0.01) for in vivo experiments. We imposed a vertical force on the tibia and solved for static equilibrium. We repeated this for different force values within the experimental range to obtain force-deformation curves.

First, we simulated an experiment from Ker et al. where they measured longitudinal elongation of the foot under vertical loading in different conditions by subsequently removing more soft tissue [2]. To replicate this experiment, we used a model in which individual ligaments were represented (before lumping them for computational efficiency in gait simulations). Ker et al. [2] did not specify whether intrinsic muscles were cut. We considered removing intrinsic muscles as an additional step after removing plantar fascia, and before removing long plantar ligament. To compare simulation outcomes to experimental data, which was obtained on the foot of an 85 kg male, we scaled forces by the ratio of body weights (0.76) and elongation by the square root of this ratio (0.87).

Next, we simulated two experiments that evaluated the effect of flexing or extending the toes on arch compression. Yawar et al. tested how extending the toes (15°) influenced the vertical compression of a cadaver foot (defined as vertical displacement of the transected tibia),

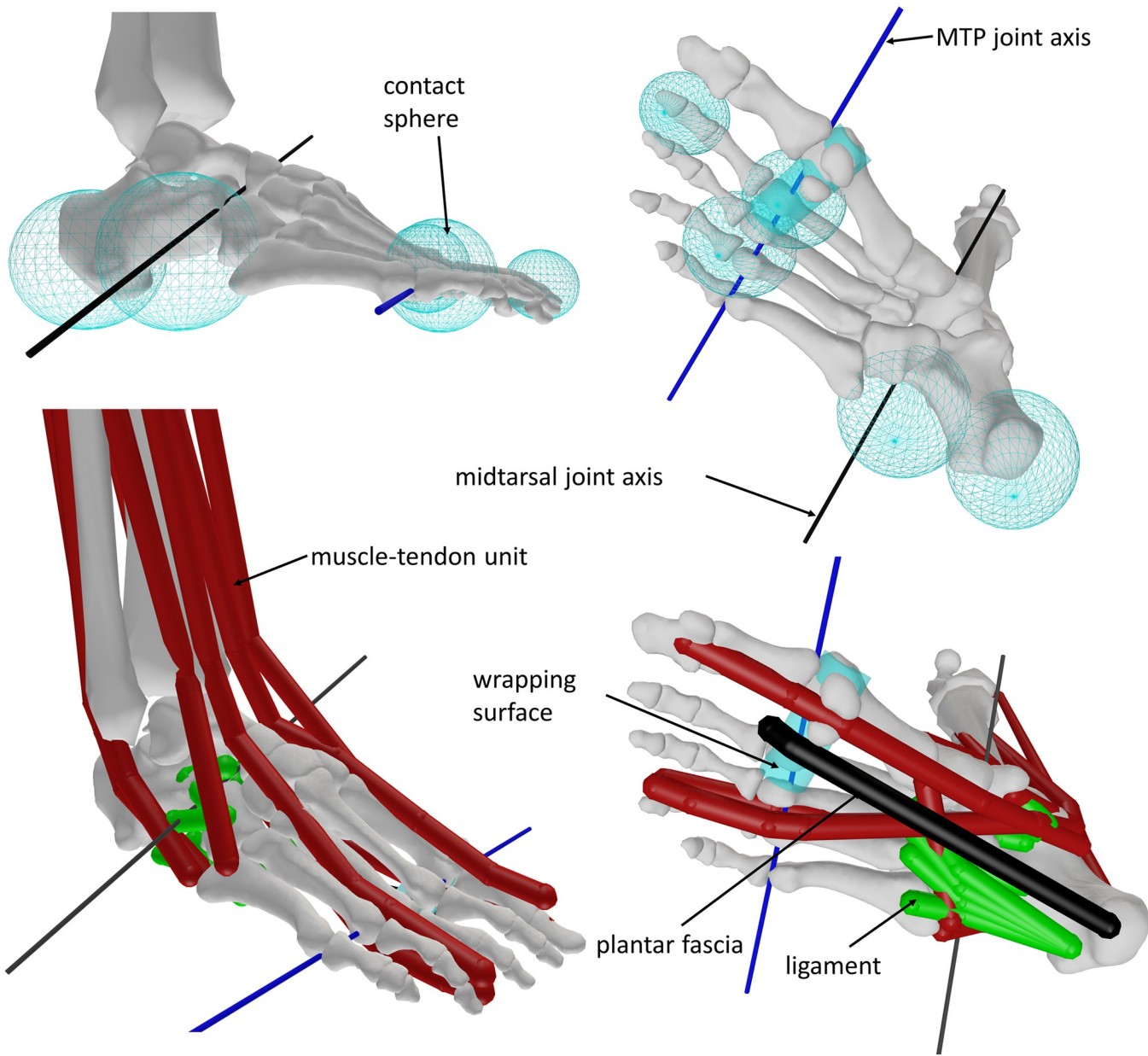

**Fig 1. Musculoskeletal model of the ankle and foot.** Plantar intrinsic muscle is not visible, because it shares the same path as the plantar fascia. Visualised via OpenSim [38,39].

compared to having the toes in their neutral position [19]. Welte et al. applied a vertical load up to body weight on the thigh of a seated person (the force was aligned to be vertical to navicular) and assessed how flexing or extending the toes by 30° influenced arch compression [18]. Arch compression was defined as the change in the vertical position of a marker on the navicular normalised to the maximal height of that marker for the subject [18].

## Simulations of walking

We used our previously developed framework [11,41] to simulate walking. We formulated gait as an optimal control problem. We solved for muscle excitations that result in a prescribed

average forward velocity of the musculoskeletal model (1.33 m s$^{-1}$, self-selected speed of the subject), and minimise an objective function. We imposed symmetry between the left and right steps, which limits the time horizon to half a gait cycle. The objective function was a weighted sum of squared metabolic energy rate, squared muscle activations, squared joint accelerations, and squared joint limit torques, integrated over the time horizon and divided by the distance travelled. We calculated the metabolic energy of muscles with the model presented by Bhargava et al. [66], which we made continuously differentiable by approximating conditional statements with a hyperbolic tangent (supplement S8). This framework was implemented in MATLAB (The Mathworks Inc., USA). We used direct collocation with a third-order Radau polynomial basis. We used algorithmic differentiation (CasADi 3.5.5 [67]) to calculate the derivatives required for gradient-based optimisation. The optimisation problems were solved with IPOPT [68] and MUMPS [69]. For a detailed description of the simulation framework, we refer to [11].

Based on convergence analysis (section 9 in in S1 Text) we selected 100 mesh intervals, cold-start initial guess and convergence tolerance of 1e-04. All results shown were obtained from simulations with these settings. To evaluate the sensitivity of predicted gait to differences in the musculoskeletal model, we performed simulations for a range of models and model parameters. All simulations were run on a laptop (allocating two threads of an intel core i7-11850 CPU to each simulation).

To evaluate the realism of our simulations, we calculated normalised cross-correlation coefficients between simulated and mean experimental joint angles, moments and powers. We considered the simulated and experimental curves to be in good agreement if the normalised cross-correlation coefficient was above 0.90 [31], and in moderate agreement if it was between 0.70 and 0.90 (cfr. threshold used in [29]). When comparing different simulation results, we used a difference of 0.05 as the threshold. We also calculated weighted root mean square errors between simulated and experimental joint angles, moments and powers as

$$\text{RMSE} = \sqrt{\frac{1}{N}\sum_{i=1}^{N}\left(\frac{\text{mean}_i - \text{simulated}_i}{\text{SD}_i}\right)^2}, \tag{1}$$

where mean$_i$ and SD$_i$ are the mean and standard deviation of the experimental data for one time point, and simulated$_i$ is the simulated value on the corresponding time point (note that simulations were deterministic and therefore had 0 standard deviation). Mesh points of the simulation were chosen as discrete time points (i.e. 200 points per gait cycle). We considered an RMSE below 3 as moderately accurate, and below 2 as accurate [29,31], and set a difference of 1 as threshold for comparisons. We made separate comparisons for the stance phase and swing phase.

To further validate our simulation results with the 4-segment foot model, we compared them to unified deformable segment analysis results published by Takahashi et al. [70]. In this analysis, deformation power distal to a reference segment is calculated based on the absolute kinematics of the reference segment and distal ground reaction forces [70]. We selected these results as reference data because they do not depend on rigid-body assumptions or predefined joints, thus better capturing the true magnitude of power [70]. To obtain simulated power distal to a segment, we summed joint powers and contact deformation powers that are distal to the segment in our model. Distal to hallux only consists of the contact sphere under the toes. Distal to forefoot includes distal to hallux and also MTP joint and contact spheres under the forefoot. Distal to hindfoot includes MTP and midtarsal joints and all contact spheres. Distal to shank is the sum of distal to hindfoot and ankle and subtalar joint powers. Work was calculated by integrating power over the stance phase.

To test whether simulations with our model can capture the effects of manipulating foot properties, we simulated walking and running with an insole that limits arch compression [6] and walking with local anaesthesia of intrinsic foot muscles [7]. To model the insole, we added a spring to the midtarsal joint, that only engages when midtarsal joint angles are positive. The properties of the insole were chosen such that they reduced arch compression by 80% during level running at 2.7 m/s in agreement with experimental data [6]. We simulated gait at 2.7 m/s for different insole stiffnesses and selected the stiffness (50 Nm/rad) that reduced arch compression by 80% relative to the same speed without the insole. We then simulated walking at 1.33 m/s with this insole. To mimic the effect of a nerve block inhibiting intrinsic foot muscle activation, we modelled that the activation should remain equal to its lower bound (0.05 cfr. [11]).

### Experimental data

We compared simulation outcomes to previously published experimental data [8]. As our simulations do not take into account uncertainty, they yielded one solution per condition (i.e. combination of model parameters and imposed gait speed) whereas we used experimental data from 10 gait cycles. The data (marker trajectories, ground reaction forces, surface electromyography) was collected from a healthy female adult (mass: 62 kg, height: 1.70 m, age: 35) during 10 gait cycles of overground walking at self-selected speed (1.33 m/s) [8]. The subject was instrumented with 59 retro-reflective skin-mounted markers. There were 10 markers mounted on each foot (supplement S10). Additionally, we measured (with measuring tape) the normal distance from the medial malleolus centre and the first metatarsal head centre to the ground while the subject was in a seated position with the foot flat on the ground. We scaled a musculoskeletal model to the anthropometry of the subject, as described in the section on the musculoskeletal model. We performed inverse kinematic and inverse dynamic analysis [38] based on the model with 4-segment feet and selected midtarsal joint orientation (Fig 1), and calculated joint powers for each individual gait cycle. After time-normalising the processed data, we calculated the mean and standard deviation at 100 points equally spread over the gait cycle duration. For the inverse dynamic analysis, we applied experimental ground reaction forces on the calcaneus, thus we did not obtain moments nor powers for the midtarsal and MTP joints. Foot joint moments are sometimes estimated by applying the total ground reaction forces to a more distal segment once the centre of pressure moves anterior to the joint [71]. This method assumes that only one foot segment is in contact with the ground at each instance in time. We did not make this assumption in our model and therefore we did not compare simulated moments to moments computed based on experimental data.

### Results

#### Simulations of standing foot under vertical loading to validate model parameters

To evaluate the realism of our foot model, we simulated a series of experimental studies that applied controlled vertical loads to the foot.

Loading the proposed foot model by a vertical force up to 3 kN resulted in a maximal longitudinal elongation of 3.5 mm (Fig 2A), which is lower than the 6–8 mm observed for an intact cadaver foot [2]. When using a more compliant plantar fascia stress-strain characteristic (Gefen et al. [50]), the simulated elongation was 5.8 mm and thus in closer agreement with experimental data. The shape of the force-elongation curve agreed with experimental data. Overall, our model captured the contribution of the different ligaments and passive muscle

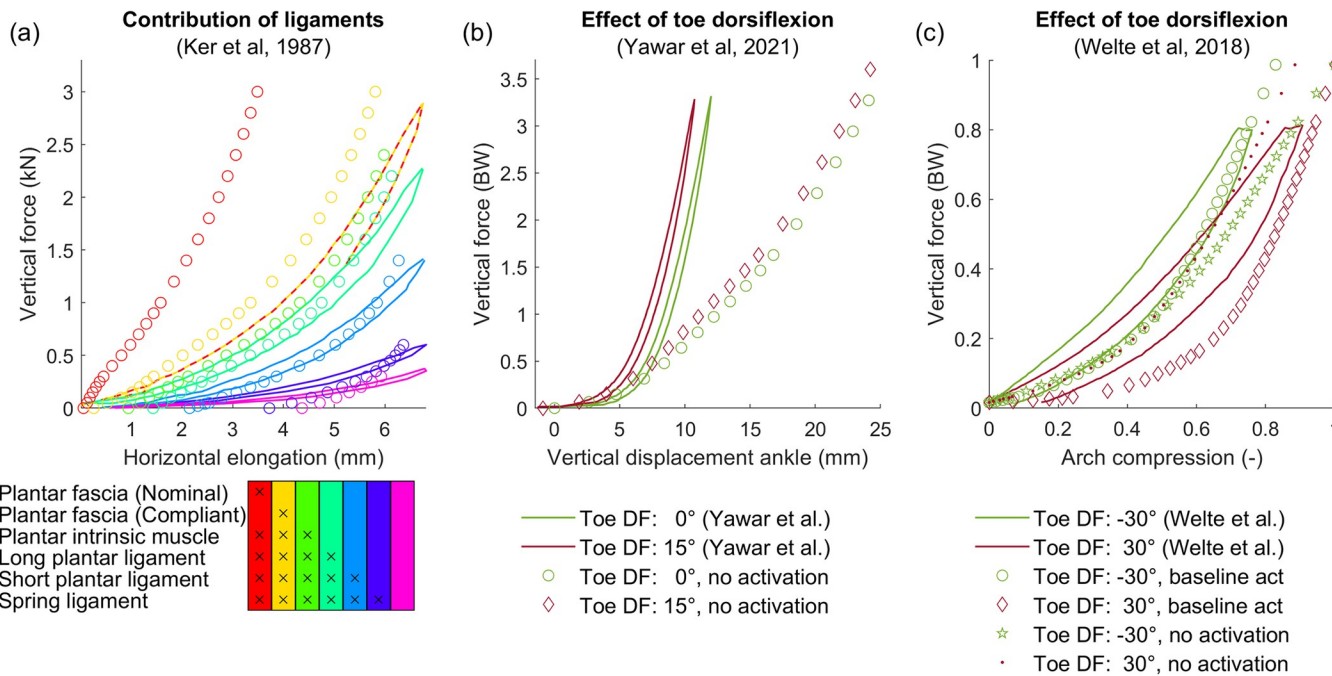

**Fig 2. Deformation of a standing foot under vertical compression.** Full lines indicate experimental data (digitised from (a) Ker et al. [2], (b) Yawar et al. [19], (c) Welte et al. [18]), and circles represent simulations. (a) Repeated loading of a cadaver foot after sequentially removing different soft tissue. The table gives an overview of structures that are present in each condition. Elongation was computed as the difference in length between calcaneus origin (defined as the most inferior, lateral point on the posterior surface of the calcaneus [73]) and MTP joint centre during the virtual experiment with respect to the resting condition, i.e. the unloaded intact foot. (b) Vertical force-displacement curves for different toe dorsiflexion (DF) positions. Displacement was calculated as the vertical position of the ankle joint centre, relative to its position when no external load is applied to the foot with toes at 0° DF. (c) Vertical loading of a standing foot of an in vivo subject, with toes passively dorsiflexed 30° and -30°. Including baseline muscle activation influenced the force-displacement curves and changed the effect of the toe position.

forces to foot stiffness as can be seen by the agreement between simulated and experimental force-elongation curves after sequentially removing soft tissues (Fig 2A).

The next set of simulations involved imposing different toe positions which have been previously shown to alter the relationship between the applied vertical load and vertical compression of the foot [18,19]. In agreement with experimental data obtained from a cadaver foot [19], extending the toes by 15° shifted the force-compression curve towards lower displacements, without affecting the shape of the curve (i.e. foot stiffness remained constant). We predicted a 1.0 mm shift in the force compression curve (Fig 2B), compared to 1.3 mm observed in experiments [19]. Although our simulations captured the observed shift, the magnitude of displacement is much larger than reported by Yawar et al. (24 mm vs. 12 mm) [19]. Further analysis of the results revealed that the heel pad of our model reached compressions up to 14 mm, which is greater than the total thickness of a human heel pad [72]. Since our model yielded realistic heel pad deformations during walking simulations, and increasing contact stiffness had only minor effects on the predicted walking pattern (see Methods and supplement S6), we did not further address the discrepancies between the experimental and simulation results for this specific cadaver experiment. In vivo, dorsiflexing the toes (30°) reduced the stiffness of the foot arch (i.e. reduced the slope of the force-compression curve) relative to when the toes are plantarflexed (30°) (Fig 2C) [18]. Our simulations captured the change in stiffness with toe flexion when we imposed a small baseline activity to the muscles (0.01)—consistent with people never fully relaxing their muscles—but not in the absence of any activity. Our simulations slightly overestimated compression, especially for smaller loads. Note that

our model does not capture the observed hysteresis throughout all experiments, because we assumed static equilibrium at every discrete load.

## Accuracy of predictive gait simulation with a 4-segment foot model

We compared simulated gait patterns using the nominal 4-segment foot model, and the previous 3-segment model (Falisse et al. [8]) to experimental data to evaluate the realism of our simulations. (Fig 3, black and orange traces). Normalised cross-correlation coefficients (NCC) and weighted root mean square errors (RMSE) between simulated and experimental gait variables (joint angles, moments, and powers) are reported in S1 Text.

**Joint kinematics.**   Using the nominal 4-segment foot model improved the overall agreement between simulated and experimental ankle and foot kinematics compared to Falisse's model. RMSE values decreased and cross-correlation coefficients increased, except for the subtalar kinematics during swing (section 11 in S1 Text). Ankle kinematics showed the largest improvement. Simulations with the nominal 4-segment foot model captured the ankle plantarflexion at initial contact, followed by dorsiflexion (Fig 3A). We predicted plantarflexion at the end of push-off but underestimated its magnitude. We captured subtalar kinematics during stance, except for the inversion at initial contact. We predicted subtalar eversion during swing, but not within the experimentally observed range. We predicted that MTP was extended at initial contact, followed by little motion during mid-stance, and a steep increase in extension during late stance. Simulations with Falisse's model did not capture these features of MTP kinematics. Simulations with the nominal 4-segment foot model predicted midtarsal kinematics that were accurate throughout the stance phase (RMSE 1.2, NCC 0.97) but deviated from experimental data for the swing phase. Possible causes for this difference are 1) the midtarsal joint axis orientation is based on the functional axis during stance, and 2) the model does not include dorsal ligaments originating proximal to the calcaneus (e.g. dorsal talonavicular ligament). Using the nominal 4-segment foot model also led to more accurate stance knee kinematics than using Falisse's model. The accuracy of the predicted hip flexion increased, while swing hip adduction accuracy decreased.

**Joint moments.**   Simulations based on the nominal 4-segment foot model resulted in similar ankle, knee and hip moments and ground reaction forces as simulations based on Falisse's model. The main difference was a decrease in knee moment accuracy (stance RMSE 7.52 vs. 5.13, swing RMSE 19.78 vs. 12.13, NCC changes < 0.05), but a better estimation of the peak knee flexion moment during push-off (Fig 3B). We also found an overestimation of the ankle plantarflexion moment during mid-stance when using the nominal 4-segment foot model (Fig 3B). We could not evaluate the realism of the predicted midtarsal and MTP moments as we did not measure ground reaction forces acting on each foot segment. Simulated ground reaction forces were similar for both foot models (Fig AE in S1 Text). Simulations based on both models predicted the M-shape pattern of the vertical ground reaction forces but displayed a steeper increase after initial contact compared to the measured ground reaction forces.

**Joint powers.**   Using the nominal 4-segment foot model instead of Falisse's model improved the prediction of the ankle joint power (Fig 3C). The peak ankle power simulated based on the four-segment foot model (~ 2 W/kg) better predicted the experimental peak ankle power (~ 2.3 W/kg) than Falisse's model (~ 1 W/kg). Both models resulted in comparable hip and knee powers (Fig AF in S1 Text). They captured the pattern of measured power but underestimated the amplitude of the peaks.

**Muscle activation.**   Using the nominal 4-segment foot model instead of Falisse's model also improved agreement between simulated and experimentally observed activity of muscles actuating the ankle and foot joints (Figs 3D and AE in S1 Text). Simulation with the 4-segment

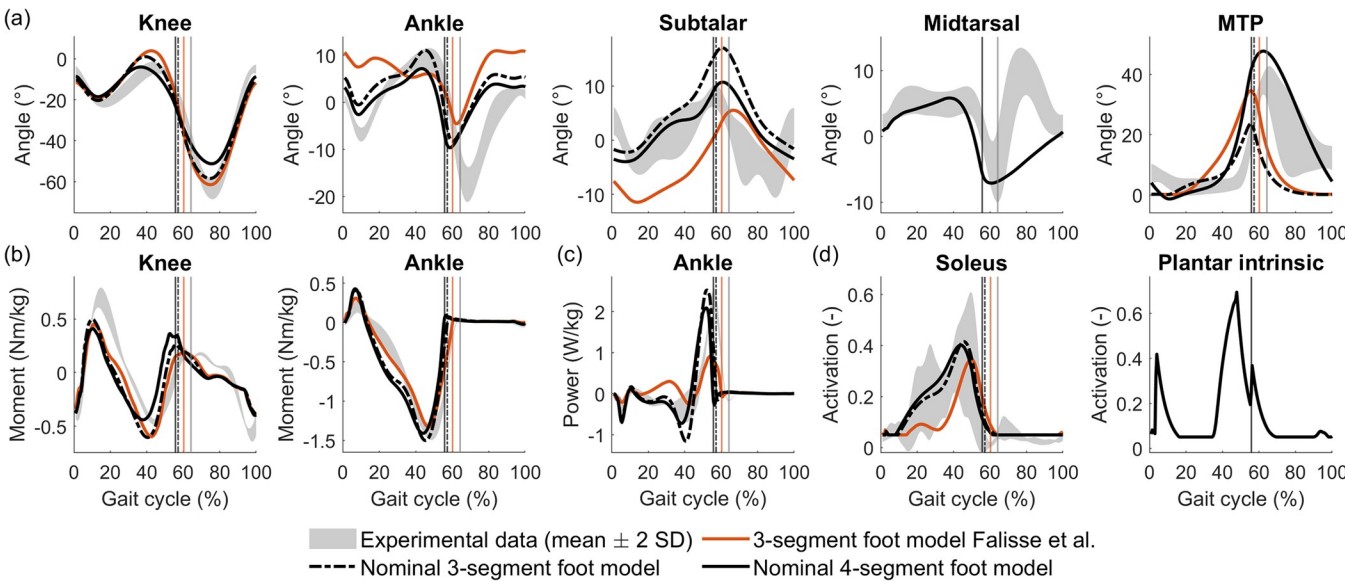

**Fig 3. Predictive simulation results with the presented model compared to experimental data and state-of-the-art simulation.** Vertical lines indicate stance-to-swing transition. (a) Kinematics. (b) Kinetics. (c) Joint powers. (d) Muscle activation. Experimental data was obtained from one subject and therefore the grey band represents stride-to-stride variability. Kinematics, kinetics, and powers of all joints can be found in section 11 in S1 Text.

foot model captured the early onset and slow build-up of soleus activity as well as peroneus longus activity. However, simulated peroneus brevis activity was inaccurate in both models. Consistent with published measurements of flexor digitorum brevis and abductor hallucis, the plantar intrinsic muscle was most active during the second half of stance [4,24] with a peak shortly before toe-off and a small peak after initial contact [23]. For both models, the simulated activity of tibialis posterior (see Fig AE in S1 Text for activation patterns of ankle-foot muscles) missed the peak during early stance that is observed in healthy adults but captures the slow build-up during mid and late stance [74]. Simulated activations of flexor digitorum and hallucis longus were in agreement with experimental data [23]. Simulated extensor hallucis longus activation captured the observed peaks in early stance and early swing, but also showed a peak during midstance that is not seen in experiments [23].

**Unified deformable segment analysis.** Foot and ankle energetics, simulated based on the 4-segment foot model, are mostly in agreement with experimental results obtained from the unified deformable segment analysis reported in the literature. Simulated power curves for different parts of the foot (i.e. distal to hallux, forefoot, hindfoot, and shank) captured the results obtained by Takahashi et al. [70], except for two features (Fig 4A). First, our simulations largely overestimated the negative power distal to shank and hindfoot after initial contact. This peak is attributed to the absorption of impact energy after heel strike [75] and coincides with the overestimation of the rise in ground reaction forces after initial contact (Fig AD in S1 Text). Second, we underestimated the negative power distal to the hindfoot, and overestimated the negative power distal to the forefoot during terminal stance (Fig 4A). Our simulation had less negative and net work distal to the hindfoot compared to experiments, but captured the observation that the net work distal to the shank is low (Fig 4B), i.e. the ankle-foot complex is close to energy-neutral during walking [70].

## Effects of modelling choices on the predicted gait pattern

We evaluated the effect of modelling choices on the predicted walking motion to explore how different foot structures shape walking mechanics and energetics. We compared the effect of

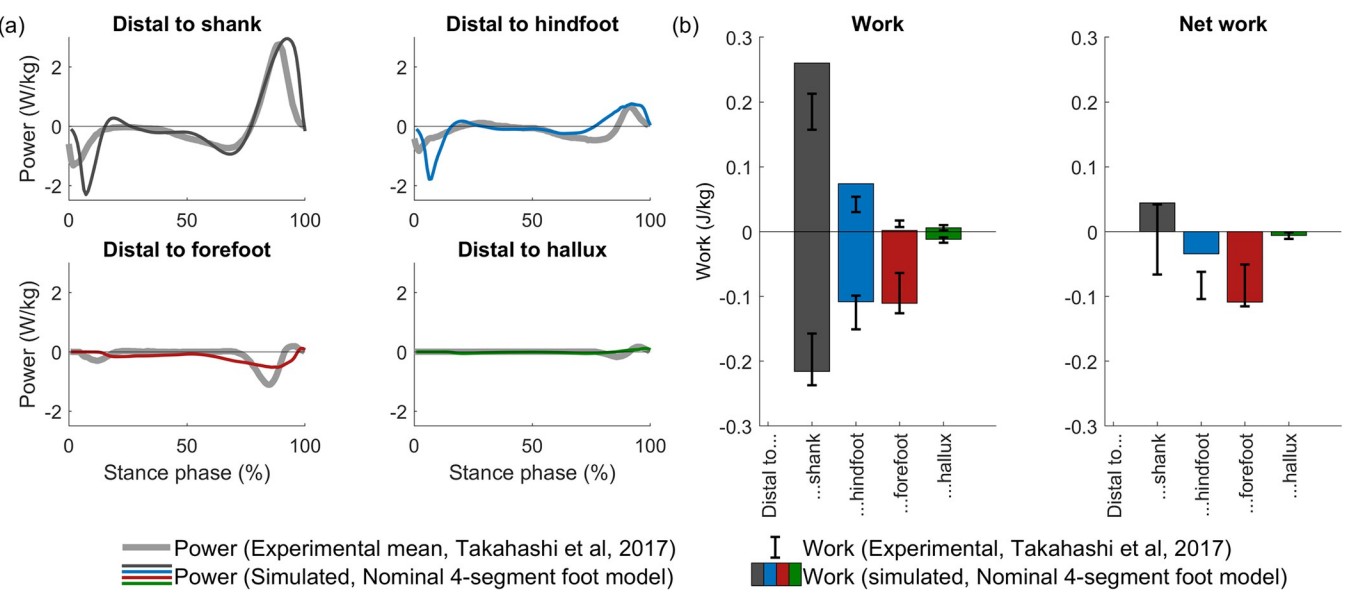

**Fig 4. Deformation power and work by soft tissue distal to ankle-foot segments.** (a) Power distal to segments. (b) Positive, negative, and net work. Experiment-based data, digitised from Takahashi et al. [70], is plotted from reference.

each specific modelling choice versus the proposed model ("nominal 4-segment foot model"). We discuss only those gait features sensitive to the change. The results of additional parameter sweeps are available in S1 Text. In all figures, black lines represent results based on the nominal model and coloured lines represent results based on a model with alternative parameter values.

**Number of foot segments.**   To dissociate the effect of modelling an additional foot segment versus altering parameters, we created a new 3-segment model ("nominal 3-segment foot model"). This model has the same complexity as Falisse's model, with parameters based on the nominal 4-segment model. Simulations based on the nominal 3-segment model captured the experimentally observed ankle dorsiflexion and soleus activation pattern better than simulations with Falisse's model (Fig 3 dotted lines, section 11 in S1 Text). Simulations based on the nominal 3-segment model overestimated the subtalar range of motion and underestimated MTP flexion at push-off. They also failed to capture peroneus longus activation (Fig AE in S1 Text). The predicted peak ankle power (~2.5 W/kg) is similar to the experimental peak (~2.3 W/kg), but the stance ankle power is less accurate than the power predicted based on the 4-segment foot (RMSE 14.27 vs. 8.53, NCC 0.74 vs. 0.83). Thus, the nominal 3 and 4-segment foot models resulted in similar accuracy for joints proximal to the ankle, but the 4-segment model better captured kinematics distal to the ankle. Given that the nominal 3-segment foot model captures ankle kinematics well, it could be considered a simpler alternative for the 4-segment model if a study does not require foot and subtalar kinematics. We did not observe a strong effect of the number of foot segments on simulation time. The gait simulation converged in 3 hours and 50 minutes when using the nominal 3-segment foot model and in 4 hours and 12 minutes when using the nominal 4-segment model. Simulations with alternative model parameters converged within 2–6 hours.

**Foot-ground contact stiffness.**   Foot-ground contact stiffness had a large influence on knee flexion and the knee extension moment during early stance when using the 4-segment foot model but not when using the nominal 3-segment model. More compliant foot-ground contact (stiffness 1 MPa instead of 10 MPa) resulted in increased stance knee extension and

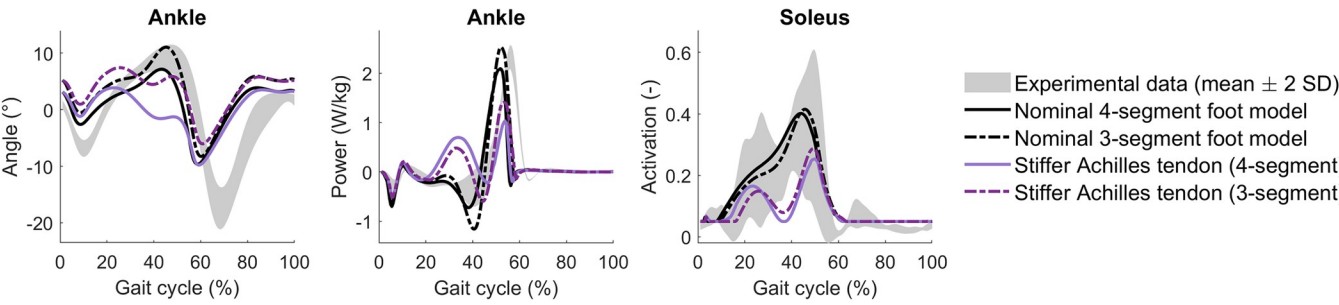

**Fig 5. Effect of Achilles tendon stiffness.** A stiff Achilles tendon results in less ankle dorsiflexion and push-off power, and a different soleus activation pattern. Experimental data was obtained from one subject and therefore the grey band represents stride-to-stride variability.

reduced stance knee extension moment in the 4-segment model (Fig Q in S1 Text). A model with stiff contact resulted in a simulated peak heel pad compression of 7.7 mm, which is still higher than the 3.8–4.8 mm reported by Gefen et al. [72]. Further increasing contact stiffness reduced compression, but did not otherwise affect the predicted gait (Fig Q in S1 Text). The contact sphere representing the heel pad dissipates 41% of its absorbed energy, which falls in the reported range of 17–44% [72,76].

**Achilles tendon stiffness.**   Achilles tendon stiffness had a large influence on ankle dorsiflexion during mid-stance. In both the nominal 4-segment and 3-segment foot models, high Achilles tendon stiffness (i.e. normalised tendon stiffness of the generic Hill-type model [32]) resulted in little or no ankle dorsiflexion during midstance in contrast to experimental observations (Fig 5). The energy storage in the Achilles tendon and ankle push-off power is lower in the simulations with a stiff Achilles tendon than in the simulations with the lower tendon stiffness that we propose here (i.e. half of the normalised tendon stiffness for the generic Hill-type model [32]). Using a higher versus lower Achilles tendon stiffness also led to a worse agreement between simulated and experimental soleus activity (Fig 5). Simulations with different Achilles tendon stiffness values showed that ankle dorsiflexion, ankle power and soleus activation peaks all increased with decreasing stiffness, while mid-stance knee extension decreased (Fig A in S1 Text).

**Plantar fascia stiffness.**   Plantar fascia stiffness had a strong influence on ankle push-off. Reducing plantar fascia stiffness (i.e. using stress-strain curves from Gefen [50] instead of Natali et al. [53]) resulted in increased ankle plantarflexion and increased midtarsal extension during stance (Fig 6A). These kinematics show similarities with pathological gait observed in patients with a midfoot break foot deformity [77]. Interestingly, reducing plantar fascia stiffness resulted in lower activation of the intrinsic foot muscle, instead of higher activation to compensate for the reduced stiffness. The muscle fibres underwent an elongation of up to 35% of optimal fibre length, which is much larger than in the nominal simulation (Fig 6D). Sweeping plantar fascia stiffness showed that midtarsal extension decreased with increasing stiffness (Fig M in S1 Text). Ankle plantarflexion and ankle power increased with plantar fascia stiffness.

**Plantar intrinsic foot muscle.**   Removing intrinsic foot muscles mainly affected the stance knee kinematics and ankle push-off power, possibly because extrinsic foot muscles that also span the ankle partially compensated for the absence of the intrinsic foot muscles to keep foot stiffness high. Removing the plantar intrinsic muscle resulted in increased stance knee flexion and earlier ankle dorsiflexion (Fig 6A). Midtarsal and MTP kinematics remained largely similar. Ankle, midtarsal, and MTP moments were higher in mid-stance, and had a lower peak in terminal stance (Fig 7A). The subtle changes in midtarsal and MTP kinematics resulted in an

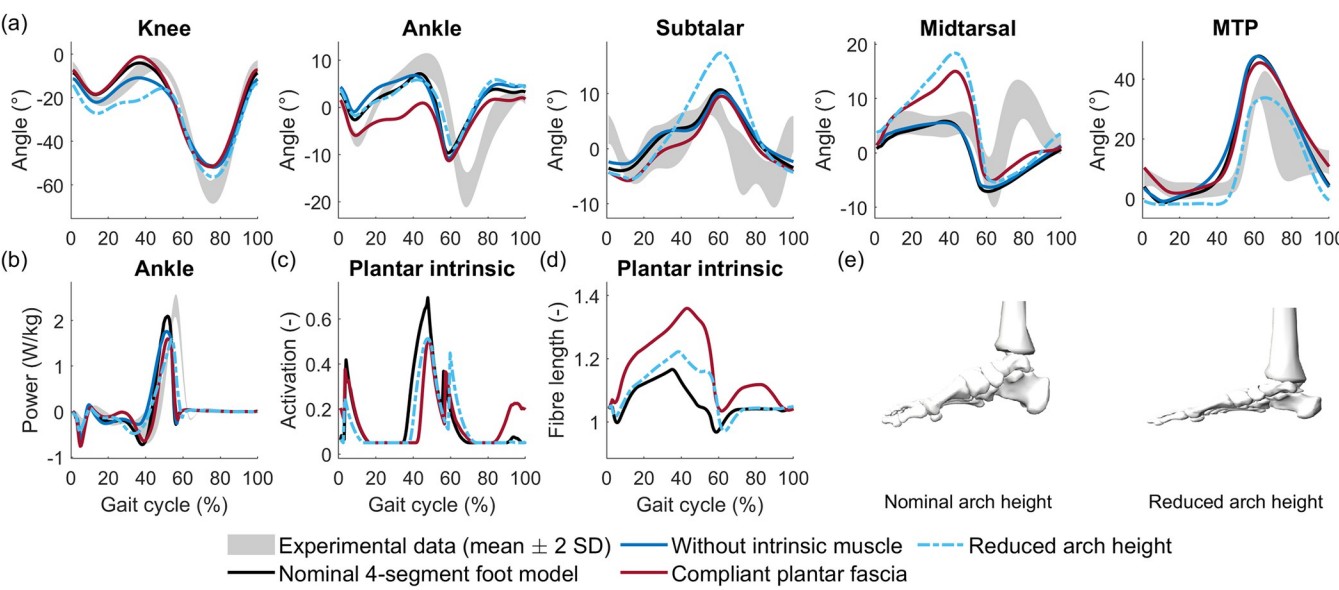

**Fig 6. Effect of foot arch stiffness.** We reduced foot arch stiffness, or the ability to stiffen the arch, by removing the plantar intrinsic foot muscle, by reducing plantar fascia stiffness, or by reducing foot arch height. All changes resulted in alterations in (a) kinematics and (b) reductions in peak ankle power. (c) Simulated activation of the plantar intrinsic foot muscle. (d) Fibre length of the plantar intrinsic foot muscle, normalised by optimal fibre length. (e) Comparison of nominal and reduced foot arch height, visualised via OpenSim [38,39]. Experimental data was obtained from one subject and therefore the grey band represents stride-to-stride variability.

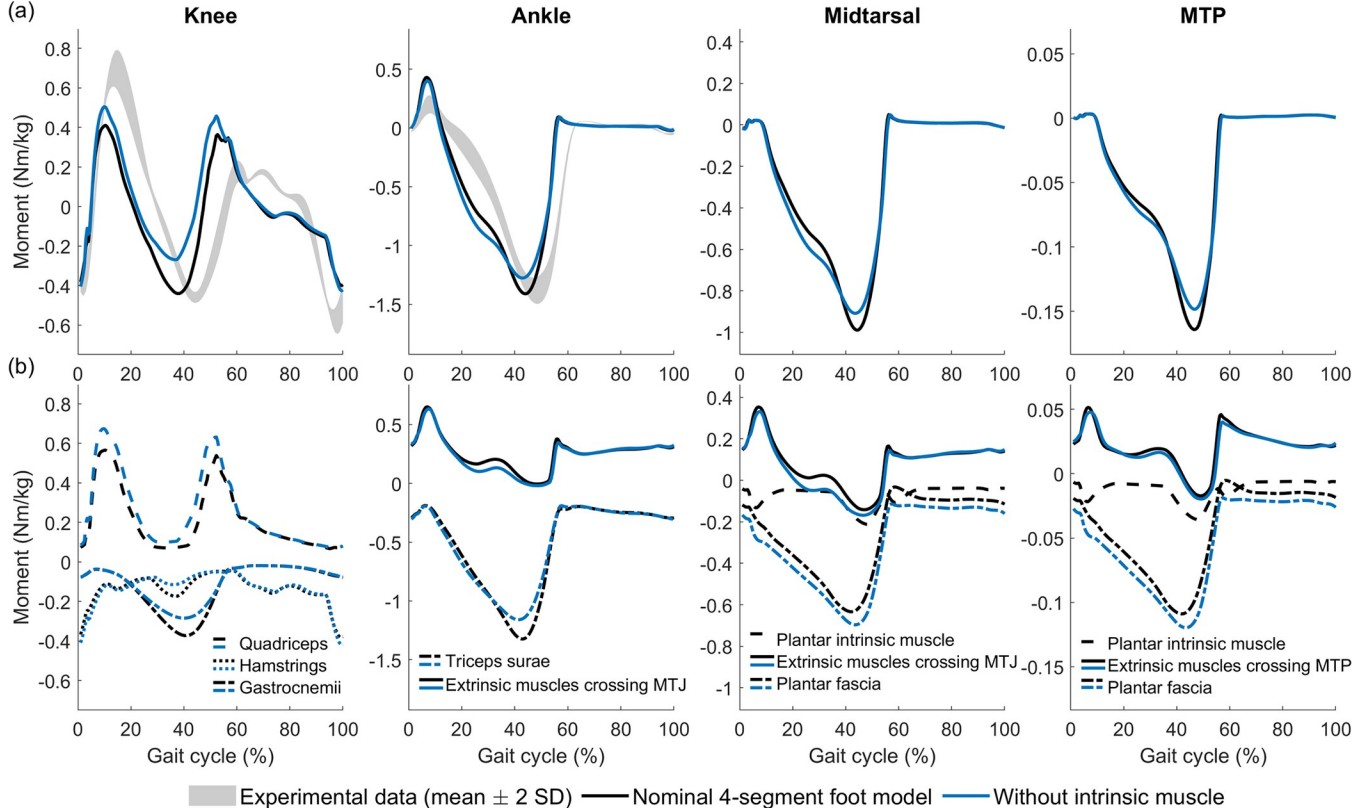

**Fig 7. Effect of removing the plantar intrinsic muscle on joint moments.** (a) Total joint moment. Experimental data was obtained from one subject and therefore the grey band represents stride-to-stride variability. (b) Joint moment by different muscle groups and plantar fascia. Extrinsic muscles crossing MTP are flexor digitorum and hallucis longus and extensor digitorum and hallucis longus. Extrinsic muscles crossing MTJ (midtarsal joint) additionally include tibialis anterior and posterior, and peroneus (brevis, longus, and tertius). Joint moments by individual muscles are shown in Fig AK in S1 Text.

increased plantar fascia strain and hence a slightly higher plantar fascia moment around both joints (Fig 7B). During mid and late stance, the midtarsal flexion moment (i.e. resisting arch compression) by extrinsic foot muscles increased. The increased contribution of the extrinsic foot muscles was also reflected in the increased ankle plantarflexion moment in mid-stance (Fig 7B). However, extrinsic muscles and plantar fascia did not fully compensate for the contributions of the intrinsic muscle to the peak midtarsal and MTP moments in the nominal simulation, and hence the net effect of removing the intrinsic foot muscles was a reduced stiffness of the foot at push off. Removing the plantar intrinsic muscle resulted in lower peak triceps surae activation (Fig AJ in S1 Text) and a lower peak ankle moment at push-off (Fig 6B). The reduction in triceps surae knee moment, combined with increased quadriceps and decreased hamstrings knee moments, resulted in a lower knee flexion moment in mid-stance, which was not in agreement with experimental results (Fig 7B). Remarkably, the simulated gait was not sensitive to parameters of the plantar intrinsic muscle. The maximal isometric force needed to be changed to at least double or half its nominal value for changes in gait kinematics to occur (Fig N in S1 Text). The optimal fibre length and tendon slack length of the plantar intrinsic muscle only influenced its normalised fibre length (Figs O and P in S1 Text).

**Foot arch height.** Simulations based on the 4-segment foot model were sensitive to foot arch height (Fig 6). Simulations with a lower arch height, which is not representative of a healthy person, resulted in a gait pattern with increased stance knee flexion and midtarsal extension. The ankle power at push-off was reduced in the simulations with a smaller arch height. The cost of transport was 6.8% higher than for the nominal model. Simulations based on the 3-segment foot model were not sensitive to the foot arch height (Fig AL in S1 Text).

## Simulating experimental manipulations to foot properties

To test whether simulations with our model can capture the effects of manipulating foot properties, we simulated walking and running with an insole that limits arch compression [6] and walking with local anaesthesia of intrinsic foot muscles [7].

**Walking and running with restricted arch compression.** Our simulations captured the effect of preventing foot arch compression on the cost of transport for walking and running. Stearne et al. created custom insoles that prevented the foot arch of their participants to compress [6]. They found that wearing these insoles did not affect the cost of transport during walking. When running at 2.7 m/s, the cost of transport increased by an average of 6% as a result of the insoles [6]. We modelled these insoles as a stiff spring on the midtarsal joint that only engages for positive flexion angles (i.e. arch compression). In the simulation, the midtarsal joint showed reduced flexion with insoles. The cost of transport barely changed for walking (+0.5%) but increased by 4.4% for running at 2.7 m/s, relative to simulating the same velocity without insole.

**Walking with local anaesthesia of intrinsic foot muscles.** Our simulations captured the effect of inhibiting intrinsic foot muscle activation on whole-body gait mechanics and energetics. Farris et al. applied a nerve block to the intrinsic foot muscles [7]. We mimicked this experiment in simulation by imposing a low constant activation to the plantar intrinsic muscles (for detailed results, see section 14 in S1 Text). Consistent with experiments [7], limiting intrinsic muscle activity in simulation hardly affected arch deformation (i.e. midtarsal joint angle) during initial loading (Fig AM in S1 Text) and cost of transport (+0.3%). Examining the contribution of each muscle to the simulated cost of transport showed that the nerve block greatly reduced the metabolic energy expenditure by plantar intrinsic muscles, but nearly every other muscle consumed more energy (Fig AN in S1 Text). Our simulations captured the experimentally observed increase in positive hip work (+6%) (Fig AO in S1 Text) and stride frequency

(+1%) and decrease in MTP quasi-stiffness (i.e. slope of line fitted between angle-moment graph) during late stance (- 6%), but underestimated the magnitude of the changes [7]. During late stance, our simulations showed an increase in midtarsal extension due to the nerve block (Fig AM in S1 Text). The corresponding increase in plantar fascia strain resulted in an 18% increase in force, which compensated for the reduction in force produced by the intrinsic muscle. In both cases, the summed force of plantar fascia and intrinsic muscle had a peak value of two times body weight. Where Farris et al. found a significant decrease in positive work done by the foot and no change in negative work [7], we predicted no change in positive work and a decrease in negative work (Fig AN in S1 Text). Notwithstanding quantitative differences, experiments and simulations both showed that inhibiting activation of intrinsic foot muscles resulted in a higher fraction of the energy absorbed by the foot being dissipated.

## Discussion

We proposed a novel 4-segment foot model for use in predictive simulations of human locomotion. Simulations based on our model generate physiologically plausible gait mechanics and energetics and elicit the effect of foot structure on whole-body gait mechanics and energetics. 1) The foot-ground contact needs to be sufficiently stiff to obtain mid-stance knee flexion. When the foot-ground contact is compliant, negative peak knee joint power is smaller, meaning that soft tissues around the knee absorb less of the impact energy from initial contact. 2) The Achilles tendon needs to be sufficiently compliant to obtain ankle dorsiflexion during stance. If the Achilles tendon is stiffer, using it to store and release energy is not the optimal strategy. 3) A stiff plantar fascia, sufficient foot arch height, and intrinsic foot muscles were important to stiffen the foot and generate high ankle push-off powers. With a reduced ability to stiffen the foot, the optimal gait strategy displayed the main kinematic features of midfoot break or crouch gait. 4) During terminal stance plantar fascia and intrinsic foot muscle transfer energy from the MTP to the midtarsal joint to further increase push-off power generated by arch recoil. Our nominal 3-segment model with well-tuned foot-ground contact and Achilles tendon stiffness also captured knee and ankle kinematics, features that had been difficult to predict with previous models, suggesting that a foot with a rigid arch and passive MTP joint is a reasonable approximation for a healthy foot during walking. However, the 3-segment model gives little insight into how different structures contribute to foot stiffness. Furthermore, a more detailed foot model is crucial to investigate how foot pathologies influence whole-body mechanics and energetics.

While simulations with the nominal 4-segment foot model resulted in physiologically plausible gait, there were still discrepancies between predicted and experimental gait. Overall, the simulated gait variables showed moderate to good shape-agreement with mean experimental data (cross-correlation coefficients > 0.70). Kinematics and kinetics of most stance leg joints showed good agreement. Most weighted RMSE values were higher than the threshold we set for good agreement (RMSE < 2). Previous simulation studies reported RMSE values below 2, but these were calculated based on data from multiple subjects [29,31]. We compared simulation results to data from a single subject and therefore the calculated RMSE values are not directly comparable to those from other studies. Intra-subject variability is smaller than inter-subject variability and therefore our standard deviations were lower, resulting in higher RMSE weights and values. Simulations with our model did capture many features of ankle-foot kinematics, energetics, and muscle activations (e.g. peak value and timing, onset of increase or decrease) that we observed in our reference data and are reported in the literature. We scaled the model to the anthropometry of the reference subject, but joint definitions, inertial parameters, and mechanical parameters of muscles, tendons, and other soft tissues are still based on an average healthy adult. Personalising the model, such as estimating subject-specific muscle-

tendon parameters [78] might improve the agreement between simulated and experimental gait for this subject (i.e. reduce RMSE) but would require more experimental data. This may not be required to study gait in healthy adults but is important when studying gait pathology for a specific patient.

Our simulations did not capture midtarsal and subtalar kinematics during swing and underestimated ankle plantarflexion angle during late stance and swing, indicating that kinematic errors are largest when forces and powers in the ankle-foot complex are low. Therefore, these errors might have little influence when using our model to study the relation between foot structure, and its contributions to load bearing and propulsion.

Our simulations showed that a stiff foot is required to efficiently generate high ankle powers. It has been suggested that the ability to stiffen the foot is important to generate a powerful push-off [17]. In simulations, we could manipulate the ability to stiffen the foot by altering arch height, plantar fascia stiffness, or the properties of intrinsic foot muscles. When we reduced the ability to stiffen the foot, simulated peak push-off ankle power decreased (Fig 5B). The magnitude and timing of negative ankle power (i.e. absorbing energy) were different for each alteration. All three changes caused the energy stored in the Achilles tendon to be approximately 18% lower than in simulations with the nominal 4-segment foot model. Our simulations suggest that if the foot cannot act as a sufficiently stiff lever, an optimal strategy is to walk with a flexed knee. This is in agreement with the hypothesised contribution of increased foot compliance to crouch gait in children with cerebral palsy [79]. An alternative strategy is to walk with excessive midtarsal extension, and to rely on elastic energy storage in plantar fascia and plantar ligaments (Fig AI in S1 Text). It is important to note that this midtarsal extension is far outside the range of motion of the data from which the functional axis was determined, thus the axis orientation might be different in a person with midfoot break. Nevertheless, these results highlight the potential of the presented model to study the relation between foot pathology and common pathological gait patterns (i.e. midfoot break [77] and crouch gait [80]).

The plantar fascia and intrinsic muscle provide push-off power around the midtarsal joint by releasing strain energy and transferring energy absorbed around the MTP joint. Recent experiments indicate that the windlass mechanism contributes to running efficiency by transferring energy from the MTP joint to the arch [81], and by modulating elastic energy storage in the arch [82]. Measuring individual contributions to energy storage and transfer by plantar fascia, intrinsic muscles, and extrinsic muscles has not been done experimentally. However, in simulations, we can calculate the exact contribution of each individual element to (the energetics of) the predicted gait pattern. In the 4-segment foot model, the plantar fascia and intrinsic muscle absorb most braking energy at the MTP joint and transfer this to the midtarsal joint (Fig 8). Strain energy released from the plantar fascia and the tendon of the intrinsic muscle, and active contraction of the intrinsic muscle provide additional power to the midtarsal joint. The 3-segment foot model also absorbs energy around the MTP but the energy is dissipated (Figs 8A and AG in S1 Text); thus, a rigid foot arch does not yield the most efficient push-off. Interestingly, the energy-efficient mechanisms that we find in walking simulations with a 4-segment foot model, are the same mechanisms as experimentally observed in running [81,82]. These model predictions should be evaluated experimentally. This example shows how the simulations can be used to generate hypotheses and inspire future experiments.

In conclusion, predictive simulations with our new 4-segment foot model capture the major mechanic and energetic features of healthy walking. Our model also reproduces the effects of experimental foot manipulations on foot and whole-body movement. Our simulations provide a complementary approach to experimental studies to gain insight into the relationship between foot structure and whole-body locomotion. Adapting the model parameters to represent a specific patient or patient group could advance our understanding of how foot

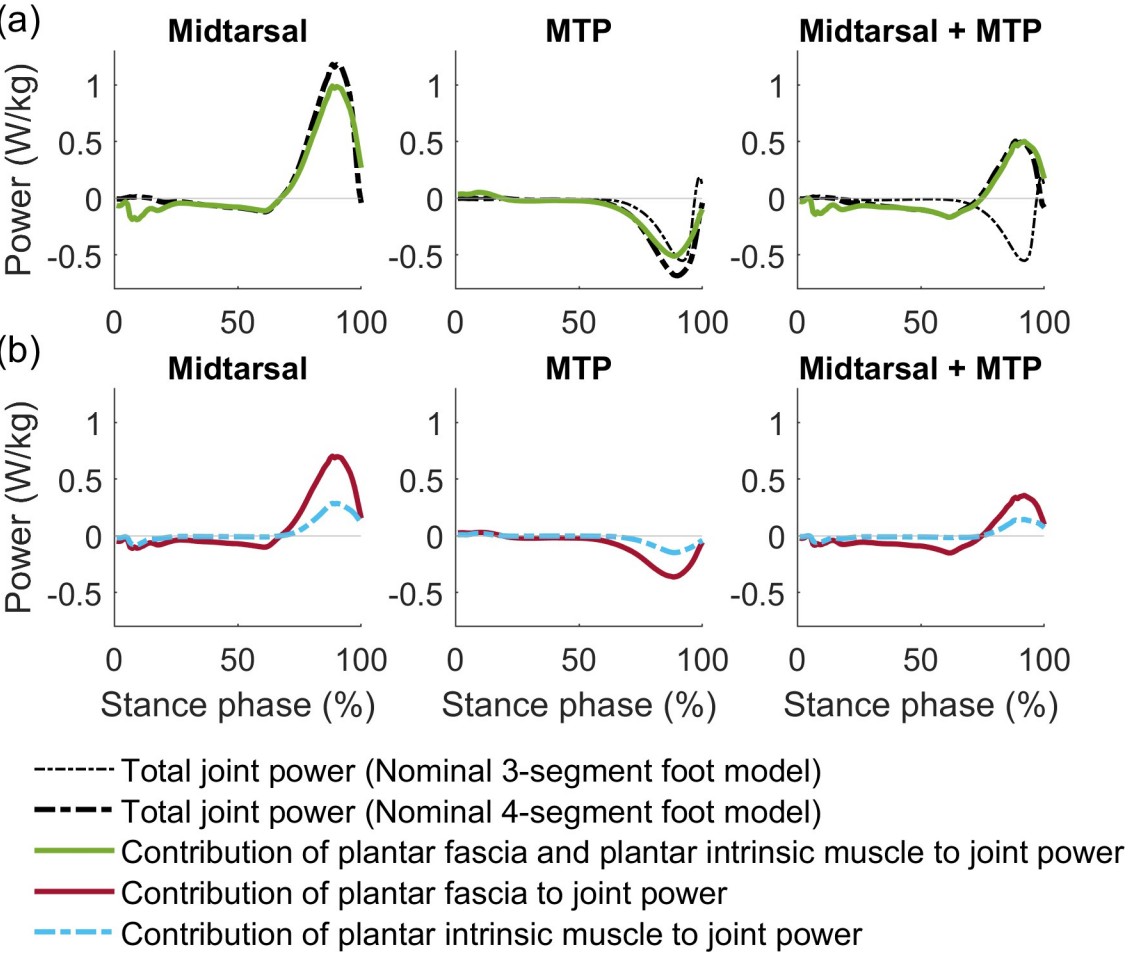

**Fig 8. Joint powers of midtarsal joint, MTP joint, and summed power of both joints.** (a) Plantar fascia and plantar intrinsic muscle account for most power around these joints. (b) Plantar intrinsic muscle power has a lower magnitude and later onset than plantar fascia power.

pathology contributes to pathological gait patterns, and these insights could eventually lead to better treatment selections and new treatments. All models and code are available at https://github.com/Lars-DHondt-KUL/3dpredictsim/tree/four-segment_foot_model.

## Supporting information

**S1 Text. Additional information about the musculoskeletal model, extended results, and sensitivity analysis.** Contains Figs A to AO and Tables A to O with their individual captions. (PDF)

## Acknowledgments

We thank Madhu Venkadesan for the useful discussions about foot mechanics and energetics and Bryce A. Killen and Hannah D. Carey for proofreading the manuscript.

## Author Contributions

**Conceptualization:** Lars D'Hondt, Friedl De Groote, Maarten Afschrift.

**Data curation:** Lars D'Hondt.

**Formal analysis:** Lars D'Hondt, Friedl De Groote, Maarten Afschrift.

**Funding acquisition:** Friedl De Groote.

**Investigation:** Lars D'Hondt, Friedl De Groote, Maarten Afschrift.

**Methodology:** Lars D'Hondt, Friedl De Groote, Maarten Afschrift.

**Project administration:** Friedl De Groote, Maarten Afschrift.

**Resources:** Friedl De Groote.

**Software:** Lars D'Hondt, Friedl De Groote, Maarten Afschrift.

**Supervision:** Friedl De Groote, Maarten Afschrift.

**Validation:** Lars D'Hondt, Friedl De Groote, Maarten Afschrift.

**Visualization:** Lars D'Hondt.

**Writing – original draft:** Lars D'Hondt, Friedl De Groote, Maarten Afschrift.

**Writing – review & editing:** Lars D'Hondt, Friedl De Groote, Maarten Afschrift.

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
