## [Decision Letter · Decision Letter 0]

9 Jan 2024

Dear Mr D'Hondt,

Thank you very much for submitting your manuscript "A dynamic foot model for predictive simulations of human gait reveals causal relations between foot structure and whole-body mechanics" for consideration at PLOS Computational Biology.

As with all papers reviewed by the journal, your manuscript was reviewed by members of the editorial board and by several independent reviewers. The reviewers were overall enthusiastic about the paper, but raised a number of issues and provided several suggestions that may improve the paper. In light of the reviews (below this email), we would like to invite the resubmission of a revised version that takes into account the reviewers' comments.

We cannot make any decision about publication until we have seen the revised manuscript and your response to the reviewers' comments. Your revised manuscript is also likely to be sent to reviewers for further evaluation.

Sincerely,

Adrian M Haith

Academic Editor

PLOS Computational Biology

Kiran Patil

Section Editor

PLOS Computational Biology

Reviewer's Responses to Questions

**Comments to the Authors:**

Reviewer #1: The aim of this work was to develop and test a 3-segment foot model that would be suitable for use in predictive simulations of human movement using musculoskeletal models. To do this, the authors worked from a previous 2-segment foot model to add a midtarsal joint (and another corresponding segment of the foot), planar fascia, and planar intrinsic foot muscles. They then performed a number of sensitivity analyses, and analyzed the changes in kinematics, kinetics, and energetics to validate the model under static loading and typical gait conditions. Finally, the authors tested if their foot model could replicate experimentally observed gait adaptations when using custom insoles or when a nerve block is applied to the intrinsic foot muscles.

Overall, the amount of work and detail in the methods are impressive. The need for a fast, but accurate foot model could have the potential for high impact for researchers performing whole-body musculoskeletal simulations. The authors have already shared models and code on GitHub. The manuscript, however, is at times difficult to follow, due to the structure, length, and the size of the figures and subpanels. A better directed Introduction could help to focus some of the needs in the Results and Discussion sections, which could both streamline the manuscript as well as provide better motivation for the reader to understand the importance of the work. More detailed comments are provided below.

Major comments:

1. Abstract: A few comments to highlight the impact of the paper better:

1a. Currently, it seems like the predictive simulations were the novel part of the work (since they are mentioned first). It could be better to highlight that a new foot was developed first, which was made to work well with predictive simulations.

1b. It is unclear why current foot models are lacking, and in which ways they don’t capture important details. The first few sentences could be shortened to more succinctly describe the shortcomings of experimental studies to make room for this.

1c. L27: Specifically mentioning which “walking mechanics” were captured would strengthen and highlight the strong validation effort.

1d. L30-L33: Is there a succinct way to highlight why replicating these results in particular opens up new opportunities for understanding foot mechanics?

2. Introduction structure: Like the abstract, the overall structure of the Introduction seems opposite of the contribution of the paper (i.e., it first discusses predictive simulations rather than the foot). The Author Summary’s structure does a better job of highlighting this. I would suggest following that model, and one way to do so would be the following:

2a. Add a new paragraph that discusses the importance of the foot, what experiments have found so far, and how models can help extend our understanding.

2b. Move the supporting information for the foot up (paragraphs that start at L92, L102, L116, L131).

2c. Discuss predictive simulations with just enough detail as needed to convey its use in the manuscript and to help support the need for a new, but still relatively few (3) segment model.

3. Introduction: There is limited discussion of previous foot models, and in particular, no discussion of more complicated models. These should be included to give a richer picture of current foot models, and a short discussion should be added as to why these are either not accurate or are not suited for your use. For instance, Maharaj et al., Modelling the complexity of the foot and ankle during human locomotion: the development and validation of a multi-segment foot model using biplanar videoradiography, specifically mentions its possible use in predictive simulations.

4. Introduction: Final paragraph. Since there was limited discussion on previous foot models (including what 2-segment models can’t capture), it’s unclear why at this point a new one is necessary. Further, this makes it more difficult to know what data will be needed to show that the new foot model sufficiently captures salient features that could not be captured before. I believe that this leads to plots with many subpanels, since the focus of the validation is unclear at this point.

5. For a paper with this much validation, I would have preferred to have the Methods section come before the Results and Discussion sections, if amenable to the journal.

5a. For example, this made it difficult to understand what the “proposed 3-segment foot model” was, when it was unclear what parameter sweeps had been performed and tested, or the differences between the 3-segment model to the two 2-segment models.

5b. L315: “curves from Gene instead of Natali et al.” were used. Unclear why one was chosen over the other if the Methods section comes after.

6. The authors should be very careful about comparison claims. They must be clearer about how comparisons were done to help the reader understand the logic, especially since no statistical tests were performed to claim any true differences. While it might be possible to do so since there were 10 gait cycles of data, it would be sufficient to more clearly state how these differences were defined. Some examples include the following:

6a. For instance, on L214 they claim that subtalar and MTP angles “clearly improved”. However, according to their tables in S10: 1) the subtalar cross-correlation in stance was lower. 2) the MTP cross-correlation in swing was lower. 3) the subtalar weighted RMSE was higher in swing.

6b. On L215, midtarsal kinematics were “very accurate”. How does this differ than just “accurate”? What thresholds were necessary? This could be helped with better information in the introduction that defines what could be a helpful level of accuracy.

6c. L224: “better estimation of the knee flexion moment during push-off and a slight overestimation of the ankle plantarflexion moment during stance”. The plots are small, so while these trends appear to be correct, it is unclear if these are meaningful differences, or if they would be within noise (especially since stride to stride variability is not described). Instances like this could be helped by reducing the number of subplots, and showing only subplots that are discussed in the main body so that they can be presented larger.

6d. In general, it was unclear why cross-correlations and RMSEs were calculated but not used as support for many points in the main text. If they aren’t used as support, it may not be necessary to report all of these. Or, if they are important to capture differences and similarities, the main text should discuss and interpret these values.

7. The authors do a good job of noting shortcomings, but often lack interpretations for how these shortcomings will affect downstream interpretations, which helps to guide readers in which use cases may be good or bad to use this model. Some examples include the following:

7a. For example, L213 “underestimated plantarflexion at the end of push-off”. How does this affect the interpretation of the results or the use of the model by others?

7b. L218: “Using the proposed model also led to slightly different estimates of hip and knee kinematics than using Falisse’s model”. Is this “different” in a good or bad way? Does this affect interpretations? If not, why?

7c. L285: “2-segment model captured the experimentally observed ankle dorsiflexion and ankle power during the stance phase but failed to capture fine details of foot kinematics”. Which details and why does this matter?

7d. L415: that cost of transport “changed for walking but increased by 10% for running”. 10% is high compared to 6% from the experiment. What were reasons why? Does this invalidate your study or why is this still good for your study?

8. Make sure that there is sufficient information in the main text itself to support points made in the main text, rather than assuming the reader has gone through the supplemental information in whole. For instance, L300: “a sufficiently compliant Achilles tendon…” is not supported by Figure 4, since it is not clear that a large parameter sweep had been performed.

9. It is unclear why the simulations of the custom insoles and of the nerve block only appear in the Discussion section. If it was difficult to decide which info should go into Results or Discussion, it may make more sense to combine the Results and Discussion sections and have subsections for validation and these other experimental tests (if amenable to the journal).

10. Make sure that each of the three models is referred to in the same way throughout the manuscript, so that the reader can better follow along. For instance, L568 describes “an alternative 2-segment foot model”, but is this the same as “new 2-segment foot model” in Fig 2? Or the “nominal 2-segment foot model” in L283?

11. In general, the figures are difficult to read due to their size, and the authors should be careful to make sure that the figures are complete. Some specific examples below:

11a. Figure 1: It’s unclear why the titles are “Foot arch stiffness”, “Midfoot stiffness”, and “Foot arch stiffness”. More detail in the caption and main text would be helpful to guide the reader through this logic.

11b. Figure 1: Is panel (b) supposed to have a y-axis of BW or kN?

11c. Figure 1: It seems like the y-axis and x-axis should be flipped (applying the force seems to be the independent variable, and displacement is the dependent variable).

11d. Figure 3: the blue and green lines are not described in the legends.

11e. All figures: the size of the subpanels are too small to see the detail that is noted in the text, often the colors are hard to differentiate, and the use of solid versus dashed lines are unclear. For instance, in Figure 5, a green line over a gray shaded region will not be color-blind friendly. In Figure S15, the color and solid/dashed line choices do not reflect the continuous numbers for the foot-ground contact stiffnesses. If the dashes were intended to help the other lines show up, it is likely better to have bigger subpanels rather than different line types switching back and forth.

11f. It may be possible to reduce the number of lines on each plot (especially in the supplemental section) by considering the minimal number of lines to convey the point that the authors are trying to make. While the authors may have run every 10% of changes between 30% and 100% (as in Figure S1), fewer lines could be plotted to support the same points, and if important to the authors to provide all of this data, could provide this as data in files rather than as plots.

12. In general, the authors are thorough, but should provide more interpretation for the reader, so that portions of the manuscript are synthesis of previous work (relevant to this manuscript) rather than a list of previous work. Some examples:

12a. For instance, L82-85: Short description of a previous 2D simulation is given, but what was both good about the simulation and what is lacking is not explicit. Providing interpretation and synthesis will better highlight the gap in the current state-of-the-art and support the current work better.

12b. L398-400: “Interestingly, the energy efficient mechanisms … are the same mechanisms as experimentally observed in running”. A little more commentary here, such as in what ways does this support the confidence and use of this model in new situations, would improve the impact for other researchers.

13. L642 states that there were 10 gait cycles of data. Were the simulation results throughout the paper an average over these gait cycles? If so, it would be good to be precise on this point.

14. A description of the simulations of the experiments with insoles and the nerve block were not included in the Methods section.

Minor comments:

1. It would be helpful if the supplemental material was arranged in a way that, when cited, increased in strictly numerical order along with the main text. It was difficult to find certain pages. This may also help if the supplemental material had page numbers on them that aligned with the table of contents.

2. L76-L78: I cannot follow along with this topic sentence. Perhaps there is a word missing?

3. L278: Could change “paragraphs” to “sections”, so that the following sections could be broken down into separate paragraphs. In particular, this could help the section “Plantar fascia and intrinsic foot model” since the different pieces of the model that were studied could be separated into separate paragraphs.

4. L341: A definition for the acronym IM is given, but not what the acronym “IM” stands for. Perhaps a symbol like “\\tilde{l}^m” could be used?

5. L374-375: Why do the simulations “confirm” in one case and “suggest” in another? It was unclear why the support for one of these statements should be stronger than the other.

6. L391: It might be better to soften language that measuring certain quantities is &

---

## [Decision Letter · Decision Letter 1]

29 Apr 2024

Dear Mr D'Hondt,

Thank you very much for submitting your manuscript "A dynamic foot model for predictive simulations of human gait reveals causal relations between foot structure and whole-body mechanics" for consideration at PLOS Computational Biology.

The revised manuscript was re-reviewed by Reviewer #1, who is largely satisfied with the revisions, but also noted a number of constructive (but minor) suggestions for further improvement, including pointing out a number of typographical errors. I am therefore asking for a minor revision to correct these errors and implement any suggestions from the reviewer that you feel would improve the paper.

Sincerely,

Adrian M Haith

Academic Editor

PLOS Computational Biology

Kiran Patil

Section Editor

PLOS Computational Biology

Reviewer's Responses to Questions

**Comments to the Authors:**

Reviewer #1: Thank you to the authors for the updated manuscript. The authors have addressed the reviewer comments well, which has made it easier to see how the work addresses needs for better foot models. Below are comments to further strengthen the manuscript.

1. How did the time to simulate the walking motions different between the different foot models? Providing this information in a sentence or two would help support that your proposed model is indeed just as tractable to use as typical musculoskeletal models with simpler feet.

2. Currently, paragraphs, especially in the introduction section, feel more like a list of literature that was reviewed rather than a synthesis of main points. Combining sentences would improve flow and readability, and help readers better understand the logic of each paragraph. A couple examples are provided below (though there are many other opportunities too):

- L97-L99: While the body moves over the standing foot, energy is stored in the Achilles tendon which rapidly releases energy to generate a powerful forefoot push-off.

- L118-121: The intrinsic muscles in the superficial layers of the plantar aspect of the foot have short fibres, a high pennation angle and a long tendon, which enables these muscles to produce force to tension their tendon to absorb and return elastic energy in a controlled way.

3. L713-715: It’s interesting that such a small change in midtarsal extension led to an 18% increase in force. However, it is difficult from the figure cited (Figure S39) to see that there’s much of a change. Can you provide a numerical value in text to make it clearer for readers?

3. Continue to review the manuscript for typographical issues. Here are some that were noted during this current review:

- L58: transfers -> transfer

- L70: do novo -> de novo

- L176: (add a comma) … designed for inverse analyses, and we believe that…

- L225: with -> by

- L254: (remove a comma) … as a single Hill-type muscle because they act…

- L307: reference issue

- L381 equation: RMSE, mean, simulated, and SD should not be italicized

- L430: we calculated mean -> we calculated the mean

- L537: Figure 3e does not exist.

- L754: Perhaps “weights -> values”?

- L770: remove the comma

- Add spaces between references to literature and figures (e.g., L222, L227, L235, L710)

- Figure S31 caption: There is a comma splice in the sentence describing the two heads of the gastrocnemius.

4. Continue to make sure that verb tenses are consistent with the context of the sentence. Some instances include the following:

- L74: were -> have been

- L76: considered -> have considered

- L77: did not consider -> have not considered

- L577: discussed -> discuss

5. Continue to review phrases in relation to comparisons being made to remove adjectives/adverbs that are not well-defined and do not help support a point or conclusion. Some examples include the following:

- L47: remove “easily”. This seems more subjective to me, and the more important point is that your work now enables other researchers (regardless of how easy it may or may not be for other researchers).

- L186: consider removing “more realistic”. The key point here is that you are validating your model.

- L531: avoid claiming “clearly a better prediction” (and in particular, the word “clearly”). Perhaps one way to reword: “The peak ankle power simulated based on the four-segment foot model (~2 W/kg) better predicted the experimental peak ankle power (~2.3 W/kg) than Falisse’s model (~1 W/kg).

- L601: remove “slightly”

- L774: remove “easily”

- L797: Consider changing “However, in simulations, this information is readily available” to “However, simulations enables us to estimate these contributions”.

**Have the authors made all data and (if applicable) computational code underlying the findings in their manuscript fully available?**

Reviewer #1: Yes

PLOS authors have the option to publish the peer review history of their article (what does this mean?). If published, this will include your full peer review and any attached files.

Reviewer #1: No

Figure Files:

Data Requirements:

Reproducibility:

References:

---

## [Editor Report · Decision Letter 2]

31 May 2024

Dear Mr D'Hondt,

We are pleased to inform you that your manuscript 'A dynamic foot model for predictive simulations of human gait reveals causal relations between foot structure and whole-body mechanics' has been provisionally accepted for publication in PLOS Computational Biology.

Best regards,

Adrian M Haith

Academic Editor

PLOS Computational Biology

Christoph Kaleta

Section Editor

PLOS Computational Biology

---

## [Editor Report · Acceptance letter]

17 Jun 2024

PCOMPBIOL-D-23-01627R2 

A dynamic foot model for predictive simulations of human gait reveals causal relations between foot structure and whole-body mechanics

Dear Dr D'Hondt,

I am pleased to inform you that your manuscript has been formally accepted for publication in PLOS Computational Biology. Your manuscript is now with our production department and you will be notified of the publication date in due course.

With kind regards,

Zsofia Freund
